# The degraded contingency test fails to detect habit induction in humans

**Sara Molinero**[1,2]*, **Pablo Martínez-López**[1,2], **Joaquín Morís**[1,2], **María J. Quintero**[1,2], **Pedro L. Cobos**[1,2], **Francisco J. López**[1,2], **David Luque**[1,2]*

**1** Department of Basic Psychology, School of Psychology, University of Málaga, Málaga, Spain, **2** The Malaga Biomedical Research Institute and Nanomedicine Platform-IBIMA, BIONAND Platform, Málaga, Spain

* sara.mol80@gmail.com (SM); david.luque@gmail.com (DL)

## Abstract

In experimental psychology and behavioral neuroscience, habits are considered stimulus-response (S-R) associations formed through extended reward training. Accordingly, habits are assessed using one of two tests: 1) Outcome devaluation, in which the value of the outcome (reward) is reduced, making it less desirable, and 2) Contingency degradation, in which the response-outcome association is reversed so that responding prevents the delivery of a reward. If a behavior is controlled by S-R links, then it should remain mostly insensitive by these two manipulations. Animal research using the outcome devaluation test has shown that initially goal-directed actions can become habitual after extended operant training. However, replicating this transition in human research has proven challenging, representing a significant problem for translational research. Notably, the contingency degradation test has rarely been used in human research. In this study, we aimed to demonstrate a shift from goal-directed to habitual control through three pre-registered experiments. Participants were trained in two S-R-O (stimulus-response-outcome) mappings for three days, with one condition (the 'overtrained') occurring four times more frequently than the other ('standard'). Importantly, we assessed the habitualization of both responses by using a degraded contingency test. Overall, we found no evidence of an overtraining effect — that is, the 'overtrained' condition did not lead to increased habitual responding. We discuss the theoretical and applied implications of these findings and explore further directions for studying habitual behavior.

Even the most skilled chef might struggle to cook a simple dish in an unfamiliar kitchen. And even if their eyes recognize the change of scenery, their hands will still instinctively attempt the same movements that worked in the old kitchen. Habits are actions we perform routinely in the same (or similar) context(s). Through repetition,

**Data availability statement:** All data used in the current study is publicly available at OSF (https://osf.io/acgx8/).

**Funding:** This work was supported by grant PID2021-126767NB-I00 (S.M, D.L.), Ministry of Science, Innovation and Universities (MICIU); grant PRE2022-103151 (P.M.L.), MICIU/ESF; and postdoctoral grant (M.J.Q), University of Málaga. There was no additional external funding received for this study.

**Competing interests:** The authors have declared that no competing interests exist.

these actions become automatic, improving efficiency and freeing up cognitive resources to attend to other tasks [1,2]. This type of habitual behavior is thought to rely on stimulus-response (S-R) associations, functioning independently of concurrent goals or motivation. Experimental studies — especially in rats — have demonstrated this using outcome devaluation or contingency degradation tests [3]. These experiments suggest that a key factor in transforming a completely goal-directed response into a more inflexible habit (performed regardless of the consequences) is the amount of experience repeating that behavior [4,5]. Less-trained responses remain sensitive to changes in outcome values or contingencies, allowing for flexible adaptation to new goals or circumstances. However, when a particular response is extensively trained, that action becomes strongly associated with the context in which it has been learned. Consequently, that response is easily triggered by contextual cues, regardless of other active goals [6].

In humans, habits have been studied across various theoretical frameworks and disciplines, generating a vast body of literature. Applied psychologists focus on characterizing habits to implement better strategies for managing healthy and unhealthy behaviors [7–9]. From a clinical perspective, an excessive reliance on habits at the expense of goal-directed strategies has been implicated in several psychopathologies, including addiction and obsessive-compulsive disorder [10–12]. Despite significant research, inducing habit formation in a controlled experimental setting has proven more challenging than initially anticipated, and researchers continue to struggle with establishing a valid and reliable procedure for experimentally inducing habits [13].

To date, most studies have used the outcome devaluation test to assess whether behavior is controlled by habits. In this approach, participants first learn to associate specific responses with rewards, typically food or points. The reward is then devalued through satiation or explicit instructions. A goal-directed strategy would lead participants to suppress or modify their responses following devaluation, whereas habitual control would be indicated by the persistence of the learned response. Using this procedure, only Tricomi et al. (2009) successfully observed habitual responses after devaluation, showing that more training led to more habitual behavior. However, replicating this finding has proven difficult, and evidence indicates that, despite overtraining, people are generally adept at adjusting their behavior during the devaluation test [13–15].

The elusive results of the outcome devaluation test in detecting habits invite us to explore an alternative—and less explored—approach within habit theory: contingency degradation. The ability of an animal to adapt its behavior to changes in response-outcome contingencies appears to depend on the amount of training [16,17], but see [18]. Theoretically, prolonged exposure to a positive R-O contingency should eventually lead to response persistence, even when the outcome is more frequently delivered when participants withhold that response. In humans, research suggests that we can accurately distinguish causal and probabilistic relationships between events, adjusting our response rates based on different positive and negative response-outcome (R-O) contingencies [19,20]. These findings have emerged from studies using minimal response training–often a single experimental session—and frequent changes in the contingencies between events. Nevertheless, to our

knowledge, the effects of overtraining in humans using a contingency degradation test remain largely unexplored (except for [21], which we will discuss further in the General Discussion).

Traditionally, based on animal research [3], it has been assumed that outcome devaluation and contingency degradation tests are equally effective for assessing habit formation in a controlled setting. However, both tests could engage the habit system through different learning mechanisms since they are supported by distinct neural circuits. For example, studies with rats have shown that lesions of hippocampal structures [22], or disruptions to specific neural connections [23,24] render animals insensitive to contingency degradation while preserving sensitivity to outcome devaluation. Even within studies that exclusively use one approach—either outcome devaluation or contingency degradation—differences in training and testing procedures are found, which rely on distinct neural pathways [25]. With this in mind, the difficulty in observing overtraining effects in humans—where extended training leads to persistent habits—using outcome devaluation tests might be overcome by employing a contingency degradation test instead.

Recently, Vaghi et al. [26] employed a contingency degradation paradigm to assess how individuals with OCD adapt their responses to changing action-outcome contingencies. Their findings revealed that while OCD patients showed intact explicit knowledge of action-outcome relationships, they responded more frequently than healthy controls in situations where an action had a weaker causal link to an outcome. These results support the idea that obsessive-compulsive symptoms may stem from an imbalance between goal-directed control and habitual behavior. However, this experiment did not investigate habit formation, as it did not manipulate the amount of training. Before using this test to compare habitual control between healthy and (sub)clinical populations, it is crucial to first establish a key prediction of habit theory: that increased training leads to reduced sensitivity to contingency degradation in healthy participants.

The current series of experiments aim to determine whether a contingency degradation procedure can succeed where other (outcome devaluation) tests have failed by demonstrating insensitivity to contingency changes and a persistent habitual response after overtraining, a key indicator of habit formation. If successful, these findings could significantly impact habit research by bridging the gap between animal models and human studies. All three experiments were pre-registered before data collection (see Method section). Conversely, a negative result would still be valuable and informative for researchers working to develop reliable experimental techniques for studying habits in the laboratory.

## Experiment 1

In this study, we aimed to provide evidence for habitual operant responses using the contingency degradation paradigm. We employed a novel design inspired by Vagui et al [26], marking the first attempt to examine the development of persistent, habitual behavior after overtraining with a positive R-O contingency.

To assess contingency sensitivity, we used the standard $\Delta P$ measure [27]. This measure represents the difference between $P1$, the conditional probability of receiving an outcome following an action, $P1 = \frac{outcome}{action}$, and $P2$, the probability of receiving an outcome in the absence of that action $P2 = \frac{outcome}{\sim action}$.

If this procedure effectively captures habitual behavior in humans, extended training with a positive contingency should promote habit formation, which might compete with goal-directed responses when the two are misaligned. Consequently, we would expect greater difficulty in adapting responses to contingency changes (from positive to negative) after overtraining.

### Method

**Transparency and openness.** All experimental studies reported here were pre-registered before data collection began. To ensure transparency and replicability, we provide detailed information on sample size determination, data exclusion criteria, experimental manipulations, study measures, and the year of data collection for each experiment [28]. All documents, data, analysis scripts, and materials employed are publicly available at https://osf.io/4tydz (Experiment 1), https://osf.io/krb42 (Experiment 2), and https://osf.io/9wguh/ (Experiment 3).

**Participants.** Because the effect that we were looking for has not been reported previously, we did not have a previous effect size to estimate the appropriate sample size. Thus, we followed the calculations of Brysbaert's [29] study, which indicates how many participants are needed to have properly powered experiments, in absence of previous similar effect. This author signaled that $d = .4$ is a good estimate of the smallest effect size of interest in psychology research and that over 50 participants are needed for a simple comparison of two within-participants conditions (in our case, standard vs overtraining) if we want to run a study with 80% power and an alpha level of .05. A total of 58 second-year university students participated in the experiment, with 52 completing all three consecutive sessions in exchange for course credits. In addition, the six highest-scoring participants at the end of the experiment received an extra payment of 25€ each. Participants were informed of this incentive before the experiment and provided written consent. The task included a manipulation check (a memory test, explained below) to ensure participants followed instructions. Only those who achieved at least 90% accuracy on this test were included in the analysis. The final sample (n = 50) consisted of 38 females and 12 males (age range = 19–34 years, $M = 20.26$, $SD = 2.74$). Participants completed the task in a quiet room with 10 semi-enclosed cubicles, each equipped with a standard PC and a 55 cm monitor positioned approximately 80 cm from the participant. This study was approved by the Human Research Ethics Advisory Panel (Psychology) of Universidad de Málaga (46–2020-H) and complied with the Helsinki Declaration. All participants reported normal or corrected-to-normal vision. Data collection took place from May 16–18, 2023 (our laboratory at full capacity allows to test up to 70 participants per day, approximately).

**Stimuli and task.** Participants were trained on several S-R-O contingencies to assess their ability to adapt their responses to subsequent contingency changes. Three figures—a circle, a square, and a triangle—were used as cues, with each figure randomly assigned to a condition (see Fig 1). In each block, one of these figures (approx. 10˚ diameter) appeared at the center of the screen in white, signaling that a response could be made. If a response was registered, the figure turned yellow to prevent multiple responses within the same 1-second time window. Each cue was associated with a specific key response, which participants were reminded of at the start of each block. Outcome

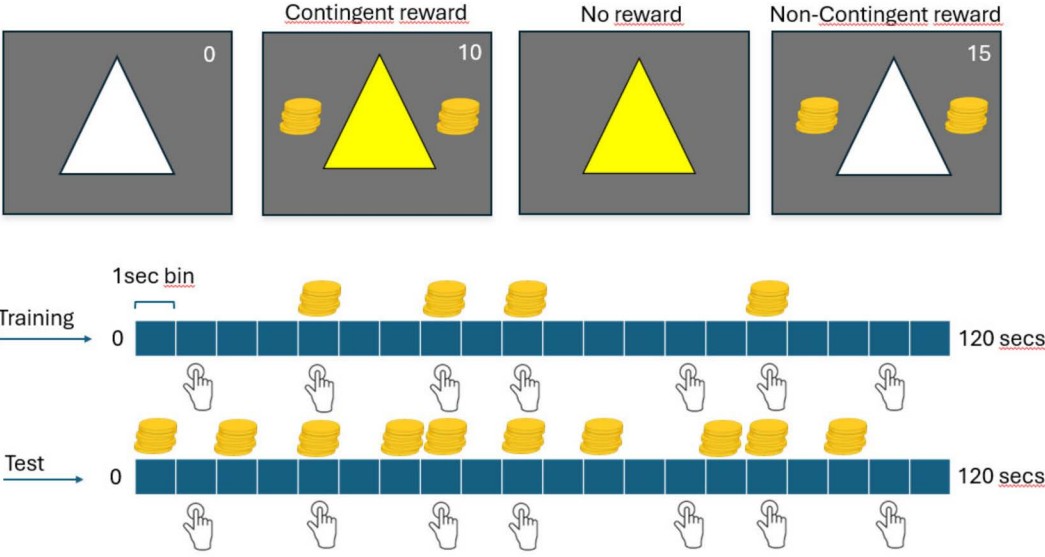

**Fig 1. The contingency degradation task.** Fig 1. Example of the stimuli displayed in the contingency degradation task. Participants were required to respond to each cue in alternating blocks of 120 secs. Positive contingency blocks were shown during training, where rewards were contingent upon making a response. Habit tests, or negative contingency blocks, were programmed by increasing the number of rewards not associated with a response, making non-responsiveness more rewarding than responding in these blocks.

delivery was represented by an image of two piles of yellow coins appearing on both sides of the geometric figure for 0.5 secs. Additionally, a continuously updated total score was displayed in the top right corner of the screen. The task was presented on a gray background and programmed using PsychoPy, version 2024.1.4 [30].

**Design and procedure.** We used a within-subject design to study the effect of overtraining on habit formation by manipulating the amount of experience with each cue, specifically by increasing the number of blocks for the overtrained cue. The experiment took place across three sessions on consecutive days. Following Vaghi et al. [26], we employed a discriminated free-operant, self-paced procedure in which participants learned and determined the response frequency that maximized their earnings in each block. Each block was divided into 120 seconds of free responding. Two different contingencies were programmed: a positive contingency $\Delta P = 0.6$ ($\frac{P1=0.6}{P2=0}$) during training (higher probability of obtaining a reward when a response was made than in the absence of a response) and a negative contingency $\Delta P = -0.3$ ($\frac{P1=0.6}{P2=0.9}$) (lower probability of obtaining a reward when a response was made than in the absence of a response) to test for habit formation (see Fig 1 and Table 1). These two contingencies were used consistently across all experiments. For brevity, we will refer to them as positive and negative contingencies.

In the first session, the three cues were presented in two separate blocks with a positive contingency. One of these cues (filler cue) was then tested in two blocks with a negative contingency. This filler cue was included to assess whether, after brief training, behavior remains goal-directed and sensitive to a sudden change in contingency. The overtrained cue was presented across all three sessions, with a total of 24 training blocks involving a positive contingency. The standard (less trained) cue was also presented across the three sessions, but only in 6 positive contingency training blocks. Both cues were tested in the third session, using two additional blocks of negative contingency for each cue. The negative contingency blocks were employed as habit tests, measuring to what extent participants can adjust their response to a change in R-O contingency [16]. At the end of each block, participants completed two tests: a causality judgment and a memory test. These tests aimed to rule out the possibility that any difference between conditions was due to a lack of attention or misunderstanding of the task. In the causality judgment, participants were asked to rate the extent to which their response produced or prevented the outcome, using a Likert scale from 100 to −100 (see specific wording in S1 Appendix, in Supporting Information S1 File). They used the mouse to select the number that best represented the contingency experienced during the last block. Following this, a memory test was administered to ensure participants had been paying attention to the cues. In this test, all cues appeared on the screen, and participants were required to identify the cue displayed during the last block by using the mouse to respond (for more details, see the pre-registration and materials employed at https://osf.io/4tydz).

**Table 1. Design and distribution of blocks in Experiment 1.**

| Session 1 (Day 1) | Session 2 (Day 2) | Session 3 (Day 3) |
|---|---|---|
| 6 × Positive (2 × O/2 × S/2 × F) 2 × Negative (2 × F) | 22 × Positive (20 × O/2 × S) | 2 × Positive (1 × O/1 × S) 2 × Negative (1 × O/1 × S) 2 × Positive (1 × O/1 × S) 2 × Negative (1 × O/1 × S) |

*Note.* The table indicates the order and number of 120-second blocks presented in each session, along with their respective contingencies: positive or negative. Negative contingency blocks are used as habit tests. O, S and F stand for overtrained, standard and filler cue, respectively. The different cues were trained in separate blocks following a pseudorandomized order to ensure they were evenly distributed throughout the session. Specifically, half of each cue type was presented in random order, followed by the second half in a new random order. For example, on the first day, the order of positive contingency blocks for a participant could be as follows: filler-standard-overtrained, and then standard-overtrained-filler. Following this, participants completed two blocks of habit tests where the filler cue maintained a negative contingency with the outcome.

Before beginning, participants were provided with detailed instructions for the task, explicitly stating that the outcome could occur even in the absence of any response. Allan and Jenkins [27] found that participants are more accurate in judging the contingency between responses and outcomes when they are aware that not responding is also a correct alternative. We made this aspect of the task explicit through the instructions, and a reminder of the instructions was presented at the start of Sessions 2 and 3. Sessions 1 and 3 each lasted approximately 25 minutes, while Session 2 lasted 50 minutes. Participants were not informed in advance about when a contingency change would occur. The order of test blocks for the standard and overtrained cues was randomized.

**Data analysis.** For the main analysis, a response ratio was calculated for each cue using the mean number of responses per 1-second bin in each block. To assess changes in behavior during the test blocks, we computed a new ratio comparing the test blocks to a baseline, which was derived from the last two training blocks (see Formula 1). If response frequency remains unchanged from training to test, this ratio will be approximately 0.5, indicating habitual responding during the test. Conversely, a decrease in responses during the test — characteristic of goal-directedness — will result in values closer to 1.

**Formula 1.** *Response ratio*:

$$\frac{\textit{Mean Responses in the last Two Training blocks}}{\textit{Mean Responses in the last Two Training blocks} + \textit{Mean Responses in Two Test blocks}}$$

Following the pre-registered analysis plan, we compared changes in the response ratio for each cue after moderate and extensive training using one-tailed *t*-tests. When necessary, *t*-tests for unequal variances were applied.

Both frequentist and Bayesian analyses were carried out using R 4.3.0 [31]. For Bayesian analysis, we used the default Cauchy prior of 0.707, a commonly used standard (e.g., JASP, 2023). Additionally, we conducted Bayes factor robustness checks in JASP 0.19.2 [32] using a wide range of possible priors (see S2 Fig in the Supporting Information).

## Results

The results displayed in Table 2 indicate that the task worked as intended, with participants experiencing contingencies closely matching the programmed ones (see also S1 Fig in the Supporting Information S1 File). This was confirmed by a

**Table 2. Contingencies, response rates, and causality ratings in Experiments 1, 2, and 3.**

| | Programmed contingency | | | Experienced contingency | | | Response rate | | | Causality ratings | | |
|---|---|---|---|---|---|---|---|---|---|---|---|---|
| | P(O\|A) | P(O\|~A) | ΔP | F | S | O | F | S | O | F | S | O |
| Exp1: Training | 0.60 | 0.00 | 0.60 | 0.60 (0.01) | 0.59 (0.01) | 0.59 (0.03) | 0.83 (0.19) | 0.88 (0.15) | 0.90 (0.14) | 46.56 (36.59) | 45.39 (35.56) | 46.68 (35.56) |
| Habit Test | 0.60 | 0.90 | −0.30 | −0.25 (0.41) | −0.26 (0.47) | −0.34 (0.48) | 0.48 (0.42) | 0.36 (0.41) | 0.38 (0.43) | −11.28 (64.98) | −28.71 (66.01) | −26.04 (67.97) |
| Exp2: Training | 0.60 | 0.00 | 0.60 | | 0.60 (0.01) | 0.59 (0.00) | | 0.88 (0.18) | 0.92 (0.12) | | 40.88 (39.38) | 39.91 (41.54) |
| Habit Test | 0.60 | 0.90 | −0.30 | | −0.15 (0.53) | −0.08 (0.51) | | 0.63 (0.44) | 0.62 (0.43) | | −4.51 (68.38) | −5.93 (66.50) |
| Exp3: Training | 0.60 | 0.00 | 0.60 | 0.59 (0.00) | 0.59 (0.02) | 0.59 (0.02) | 0.89 (0.16) | 0.91 (0.13) | 0.91 (0.11) | 26.76 (45.09) | 32.71 (43.53) | 34.26 (43.85) |
| Habit Test | 0.60 | 0.90 | −0.30 | −0.12 (0.42) | −0.25 (0.50) | −0.25 (0.51) | 0.62 (0.39) | 0.38 (0.43) | 0.40 (0.43) | −17.80 (57.34) | −28.85 (60.21) | −27.31 (61.03) |

*Note.* Programmed contingency (ΔP) is the result of subtracting P(O\|A) and P(O\|~A). F, S, and O denote filler, standard, and overtrained cues, respectively. Experienced contingency, response rate, and causality ratings data correspond to mean (*SD*). Experiment 2 did not include a filler cue.

correlation analysis between programmed delta P and experienced delta P, $r = .867$, $p < .001$. As it can be seen in S1 Fig in S1 File and Table 2, mean Causality ratings suggested that participants, as a group, were aware of contingency shifts, correctly identifying changes from positive to negative contingencies. However, exploratory individual-level analysis of these ratings showed that there were wide differences between participants in their perception of the new contingencies (see more details in S3,S4 Figs, S1,S2 Tables in the supporting information S1 File and General Discussion).

As expected, the response ratio analysis during the habit tests showed that participants were sensitive to contingency degradation both at the end of the first day and after limited training with the filler, $t(49) = 6.79$, $p < .001$, $d_z = 0.97$, and the standard cue, $t(49) = 10.20$, $p < .001$, $d_z = 1.46$, on the third day. Bayesian factor analysis provided strong evidence for these effects ($BF_{10} > 100$).

However, contrary to our predictions, sensitivity to contingency degradation was similar for standard and overtrained cues [$t(49) = -0.08$, $p = .53$, $d_z = -0.01$, $BF_{01} = 1/0.14 = 7.14$], indicating that the null hypothesis was seven times more probable than the alternative. Additionally, we compared the response ratio of the filler cue (tested during the first session) with that of the overtrained cue (tested during the last session). Once again, we found no significant differences in the habit test between limited and extensive training conditions [$t(49) = -2.33$, $p = .98$, $d_z = -0.33$, $BF_{01} = 1/0.04 = 25$], providing strong evidence in favor of the null hypothesis.

These findings do not support our main hypothesis that extended training might hinder the ability of participants to adapt their behavior to contingency changes. Instead, participants demonstrated a clear ability to judge and adjust their responses according to task demands, showing no evidence of habit formation.

**Exploratory analysis.** In contrast to session 1 and 2, we alternated blocks with positive and negative contingency during session 3 (see Table 1). This aspect of our design could have encouraged participants to keep goal-directed control during that session, improving their capacity to hold their responses during negative blocks. To rule out the possibility that our results were influenced by participants completing two separate blocks with negative contingency for standard cues and two additional blocks for overtrained cues, we conducted an additional analysis. Specifically, we examined whether experiencing a second negative contingency block (habit test) improved participants' sensitivity to contingency changes. For this, we compared performance between participants considering only the first habit test, analyzing standard cues (n = 27) and overtrained cues (n = 23). Although the difference was not significant [$t(48) = 1.572$, $p = .062$, $d = 0.46$, $BF_{10} = 1.39$], there was a trend suggesting that responses to overtrained cues were more habitual than those to standard cues (see Fig 2). However, since this analysis was not part of our initial plan, these findings should be interpreted with caution.

## Experiment 2

In the second experiment, we aimed to eliminate potential confounding effects from conducting a habit test before the final habit test. To address this, we made several modifications to our procedure. First, we removed all trials involving the filler cue to prevent any habit test from occurring in the first session. Additionally, we only conducted habit tests during the third session, with a counterbalanced order (see next section). With these changes, we expected our primary measure —the response ratio— to be more sensitive in detecting persistent responses despite contingency degradation.

### Method

**Participants.** Because we did not obtain a significant effect of overtraining in Experiment 1, we followed the same rationale as in Experiment 1 to determine sample size. Thus, we aimed for a total sample size of at least 50 participants, following Brysbaert´s [29] recommendations. Fifty-seven second-year university students participated in this experiment, completing all three sessions in exchange for course credits (44 females, 13 males; age range:19–54 years, $M = 20.98$, $SD = 5.42$). All participants achieved at least 90% accuracy on the memory test. The six participants with the highest scores received an additional payment of 25€. No participants were excluded based on memory test performance. The

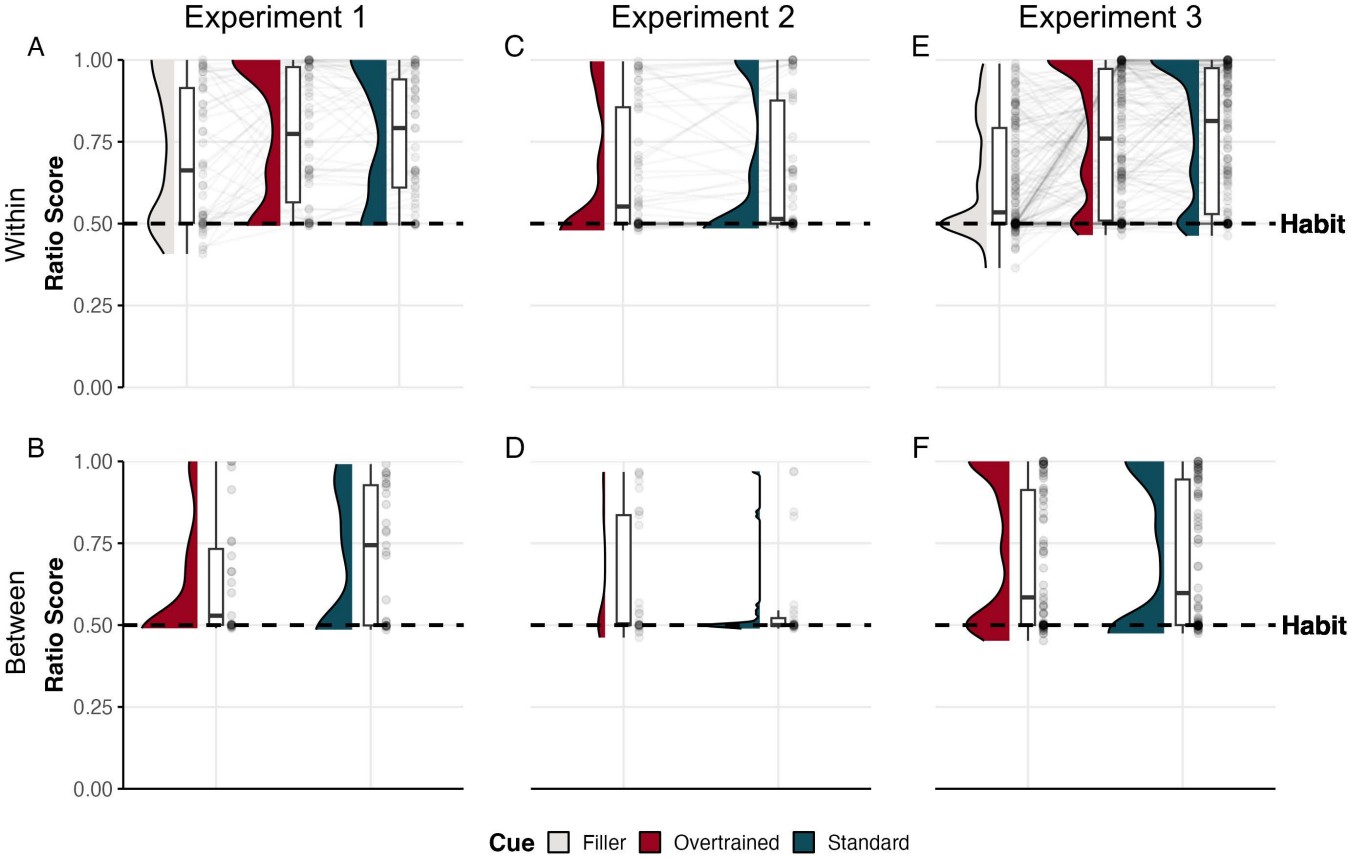

**Fig 2. Ratio score for overtrained, standard, and filler cues in all experiments.** *Note.* The ratio scores for each experiment during the test phase are displayed in columns. Within-subject (A, C, E) and between-subject comparisons (B, D, F) are shown in rows. According to the formula applied to calculate the ratio, scores closer to 1 indicate goal-directed behavior, whereas scores closer to 0.5 reflect habitual behavior (persistence of response despite the negative contingency).

experimental setting and equipment employed were identical to those described in Experiment 1. Data collection took place from May 23–25, 2023.

**Design and procedure.** The distribution of blocks was as follows: during the first session, participants completed 4 blocks with positive contingency (2 standard/2 overtrained) and 22 blocks in the second session (2 standard/20 overtrained), also with positive contingency. Block order was pseudorandomized. Half of the blocks for each cue type were presented in a random order in the first part of the session, followed by a new random order in the second half. Session 3 included both training and habit test blocks, following an alternating sequence: two positive contingency blocks→two habit test blocks→two positive contingency blocks→two habit test blocks. The test order for the standard and overtrained cues was counterbalanced across participants (whereas in Experiment 1, the order was fully randomized, leading to unequal sample sizes between groups).

Additionally, we refined the wording of the causal rating question to ensure participants' judgments reflected the perceived contingency in each specific block. We added the phrase: *"Based on the last two minutes, …"* before the causal question. All other task parameters remained identical to Experiment 1.

**Data analysis.** For between-subject comparisons involving only the first test block, we calculated the response ratio using a single block with positive contingency as the baseline — specifically, the last block of training before the test. The remaining analyses followed the same approach as in Experiment 1.

## Results

Consistent with the findings from Experiment 1, participants successfully reduced their response rates during negative contingency blocks. This was evident in the analysis of the standard cue, $t(56) = 5.90$, $p < .001$, $d_z = 0.79$, with strong evidence ($BF_{10} > 100$) supporting this effect. However, contrary to our predictions, we found no evidence that extended training led to insensitivity to contingency degradation. This was true both in a within-subject comparison using the same response ratio as in Experiment 1 ($t(56) = 0.16$, $p = .43$, $dz = 0.02$, $BF_{01} = 6.25$) and in a between-subject comparison (standard cue: $n = 28$; overtrained cue: $n = 29$) using the new ratio with a one-block baseline ($t(55) = -1.457$, $p = .92$, $d = -0.40$, $BF_{01} = 8.33$). In both cases, the null hypothesis was significantly more probable than the alternative. These results cannot be attributed to task comprehension issues or discrepancies between the experienced and programmed contingencies, as indicated by a strong correlation between the two ($r = .790$, $p < .001$).

## Experiment 3

In Experiments 1 and 2, we found no evidence of habit induction, as participants' performance in the test blocks remained unaffected by the amount of prior operant training. However, an exploratory analysis in Experiment 1 suggested a trend toward more habitual responses (ratio score close to 0.5) with overtrained cues than with standard cues (see Fig 2). This trend could either be statistical noise or a genuine overtraining effect indicative of a new habit. Because overtraining effects have not been previously reported using a contingency degradation protocol, we did not have enough data to calculate an appropriate power analysis for Experiment 1 and 2. That means that the trend observed in Experiment 1 could have been significant with a larger sample size. To investigate this possibility further, we conducted Experiment 3, a higher-powered replication of Experiment 1. Notably, we pre-registered the between-subject analysis that previously showed a promising trend in Experiment 1.

### Method

**Participants.** The between-subject comparison in Experiment 1 revealed a non-significant trend. To achieve a similar effect with a statistical power of .80, we used G*Power 3.1 [33] to estimate the necessary sample size. The calculation indicated a requirement of at least 62 participants per group, totaling 124 participants. A total of 151 first-year university students enrolled in the study in exchange for course credits and a potential monetary reward, as in previous experiments. Of these, 147 participants completed all sessions (116 females, 29 males, 2 non-binary; age range: 17–47 years, $M = 19.50$, $SD = 4.27$). No participants failed the memory test criteria. Data collection took place between October 24 and November 16, 2023.

**Design and procedure.** The design and procedure were identical to those used in Experiment 1, except that the test order in the third session was counterbalanced instead of randomized. This ensured an even distribution of participants across groups in the between-subject comparison. Additionally, we retained the improved wording for causality ratings introduced in Experiment 2.

**Data analysis.** Within-subject and between-subject comparisons followed the same analytical approach described in Experiments 1 and 2.

### Results

Consistent with our previous experiments, participants showed sensitivity to contingency changes after limited training with the filler, $t(146) = 9.42$, $p < .001$, $d_z = 0.78$, $BF_{10} > 100$, and standard cues, $t(146) = 16.37$, $p < .001$, $d_z = 1.35$, $BF_{10} > 100$. Causality ratings also confirmed that participants accurately perceived these changes in contingency relationships. Moreover, we observed a strong positive correlation between experienced and programmed contingencies, $r = .833$, $p < .001$. However, as in Experiments 1 and 2, we found no evidence of habit induction following overtraining. The within-subject analysis showed anecdotal evidence for the null hypothesis when comparing the overtrained cue with the standard cue

[$t(146)$ = 1.31, $p$ = .096, $d_z$ = 0.11, $BF_{01}$ = 1/0.38 = 2.63] and very strong evidence when is compared with the filler cue [$t(146)$ = −6.98, $p$ = 1, $d_z$ = −0.58, $BF_{01}$ = 1/0.01 = 100]. Nevertheless, our primary focus was the between-subject comparison between the first standard and overtrained habit test block. This analysis also yielded null results [$t(145)$ = 0.30, $p$ = .38, $d$ = 0.05, $BF_{01}$ = 1/0.22 = 4.54], indicating that the null hypothesis was four times more probable than the alternative.

## General discussion

The transition from goal-directed responses to automatic, habitual behavior is thought to depend on the amount of experience in a specific context. As exposure increases, S-R associations are strengthened, dissociating the action from its outcome representation [2]. Once the habit has developed, associated cues may trigger the overtrained response even when it no longer leads to the desired outcome [5,16]. Animal research has demonstrated habit formation using outcome devaluation and contingency degradation tests, with habitual behavior emerging as a function of training [3]. Human studies have attempted to replicate this pattern of results by using the outcome devaluation test, but the findings have been inconsistent [13–15,34,35]. Surprisingly, few studies have assessed whether contingency degradation tests can effectively track habit development in humans. The present experiments aimed to address this gap, providing knowledge of how humans respond to contingency degradation following varying amounts of operant training.

In three pre-registered experiments, we found no evidence that extended training led to habitual responding in a contingency degradation test. Despite three days of overtraining under a positive contingency (where pressing a key led to an increase in the probability of the reward), participants reduced their response rate just as much as those in a less-trained control condition when the contingency shifted to negative (where pressing the key reduced reward probability). We manipulated training duration within subjects by increasing the number of training blocks for one S-R-O mapping—a strategy commonly used in human habit research (e.g., [36]). Importantly, the same null effect was observed in between-subject comparisons, even when analyzing only the first test block. This rules out the possibility that experiencing multiple test blocks increased sensitivity to contingency changes and masked habit expression.

Importantly, causality ratings and memory test results indicated that most participants understood the task requirements and accurately detected changes in contingency relationships. Thus, our findings suggest that participants' behavior in this task remained goal-directed, regardless of the amount of prior operant training.

An alternative explanation of these null results is that, in this paradigm, habits might develop even after short training, making it difficult to detect differences with overtraining conditions. This possibility seems to be supported by the data from some of the standard cues during the habit test. For instance, we can see some participants with ratio scores very close to 0.5 in Experiment 2, even for the standard cue (see Fig. 2). Although plausible, there might be a simpler explanation; some participants might not be as good at detecting contingency changes as what is indicated by level group analysis. Obviously, if a participant is still under the impression that the response is a good way to get the reward, they will continue responding—but these 'perseverant' responses are goal-directed actions and not habits. For assessing this possibility, we analyzed the correlation between ratio score and causality ratings during blocks with negative contingency (habit tests). As we suspected, we obtained a significant negative correlation in all experiments for all cues (see S3,S4 Figs and S1,S2 Table in S1 File), indicating that those participants with higher perception of causality in negative blocks—they still believe that their behavior cause the outcome—also showed lower punctuations in ratio score, that is, closer to 0.5 (indicative of habits). However, those seemly habitual responses from participants who are not aware that the contingency changed are in fact goal-directed actions. Interestingly, similar results have been observed in outcome devaluation protocols: that is, seemingly habitual responses which are the consequence of suboptimal outcome devaluation at least for some participants [35,37]. All these results highlight the need for a careful examination of the functioning of devaluation/degradation protocols at the individual level before making claims of evidence for habits, especially after short training.

Overall, our results do not support the validity of the methodological approach used in these experiments for studying habit induction in humans. This has implications for previous research that has employed similar designs —namely,

a free-operant training stage followed by a contingency degradation test— to investigate habit formation. For example, Vaghi et al. [26], following the approach of Liljeholm et al. [38], used this paradigm to compare performance between OCD patients and healthy controls. They found that OCD patients exhibited greater response persistence during degraded contingencies than controls despite both groups being equally aware of the contingency changes. The authors concluded that OCD patients show a dissociation between action-outcome knowledge and habitual responding, suggesting a dysregulation between goal-directed and habit systems in OCD. This aligns with broader theories proposing that transdiagnostic impulsivity and compulsivity stem, at least in part, from an imbalance between these two systems ( [11,12,39,40]; but see [41] or [42]).

Our experiments differed from previous studies using contingency degradation tests in several ways. Prior research [21,26] introduced frequent shifts in response-outcome contingencies. For instance, Vagui et al. employed twelve 2-min blocks with different contingency relationships in a single 34-minute session, covering six $\Delta P$ conditions (two positive, two negative, and two zero-contingency). However, habit learning is known to require stable contexts [43], and frequent contingency changes could interfere with this process. To address this, we deliberately minimized contingency shifts, implementing only a single change in the between-subjects analysis to better facilitate habit formation.

Our experiments also differ from previous studies in their interpretation of persistent responding when a positive contingency is weakened but remains positive (for instance, from $\Delta P = 0.6$ to $\Delta P = 0.3$), assuming that this constitutes evidence for habitual control of behavior. However, there are reasons to be cautious about that assumption. When the cost of responding is zero, maintaining a high response rate remains the optimal strategy for the participant as long as $\Delta P$ is positive—even if it has recently transitioned from a stronger positive contingency. In other words, persistent responding when $\Delta P$ is reduced but still positive may reflect goal-directed behavior aimed at maximizing rewards. In studies such as Vagui et al. [26], more convincing evidence for dysregulated habit expression in OCD would have been a higher response rate during negative contingency blocks compared to healthy controls. However, OCD patients primarily showed increased responding during low-but-positive contingency blocks—an approach still aligned with goal-directed reward maximization. To avoid this potential confound, our habit tests exclusively used blocks where contingency was reduced below zero, ensuring that continued responding was objectively counterproductive for participants.

Recently, Nebe et al. [21] tested participants using a modified version of the contingency degradation procedure from Vaghi et al. [26], extending the training to four days. However, they found no increased insensitivity to contingency changes between the first and last training sessions in a within-subject comparison. In other words, similar to our findings, they failed to observe habit expression as a function of the amount of training. While Nebe et al.'s study is informative, it shares a key limitation with previous research: as in Vaghi et al. [26], contingency relationships changed frequently throughout training. Such conditions likely encouraged participants to maintain goal-directed control over their behavior, potentially explaining the lack of a clear effect of overtraining on habit expression.

We believe that our methodological approach provides a more valid framework for studying habit formation. By maintaining stable contingencies, we aimed to foster habit development, and by testing habits under negative contingencies, we minimized potential confounds with goal-directed behavior. To assess the effect of training duration, we conducted both within- and between-subject comparisons. Each participant completed two contingency degradation test blocks—one for the overtrained cue and another for the standard cue—allowing us to directly compare the effects of training within subjects. Additionally, to isolate the impact of training between subjects, we analyzed only the first test block for each participant. This ensured that participants in the between-subjects analysis had not encountered any contingency shifts before testing, maintaining a stable learning environment across the three training days.

Our learning task allowed participants to reach a high-performance (asymptotic) level. Causality ratings during training and test blocks were highly accurate, confirming that participants learned all the contingencies and recognized the changes between test conditions. Importantly, ratings during test blocks were, on average, negative, indicating that participants understood that responding reduced the likelihood of receiving a reward (see Table 2). Participants' operant

responses were aligned with these ratings, as they consistently reduced response rates for both overtrained and standard cues. Crucially, this reduction was of the same magnitude across conditions, strongly suggesting that the few responses observed during the contingency degradation tests were not true habits but rather goal-directed actions—possibly driven by boredom or lack of attention—as suggested by the differences in their perception of the negative contingencies (see S3,S4 Figs and S1,S2 Tables in S1 File).

One potential counterargument is that the induced habit was not strong enough to be detected in the contingency degradation test after three days of training. While we cannot determine with certainty what would have occurred with extended training, several factors suggest that our findings would remain unchanged. First, response rates for the overtrained cue reached an asymptotic level after just three blocks (see S1 Fig, in Supporting Information S1 File). Second, previous research has demonstrated that habit expression does not markedly increase when training is extended from a few days to several weeks [44]. Third, analyses of response times in goal-directed tasks suggest that S-R habits can form within three days of training, though not all measures are sensitive enough to detect them [45]. Taken together, these observations suggest that the contingency degradation test lacked the sensitivity required to detect habits formed over three days of training. However, other work has reported high variability in participants regarding the amount of practice necessary to develop habitual behavior [46], so it would remain possible that contingency degradation paradigms might need more extensive training to detect overtraining effects.

Thus, our results suggest that the contingency degradation test is not a more effective strategy for studying habit formation than the outcome devaluation procedure, at least with the parameters that we have employed. Regardless of the chosen method, detecting habit induction as a function of training duration remains challenging, which raises concerns about the validity of these measures. Recent research has attempted to address this issue, identifying key factors influencing habit expression. One such factor is whether goal-directed control during testing involves actively responding or withholding a habitual response. Du and Haith [47] demonstrated that when the decision is simply to respond or not, it remains easily controllable and may require an extensive amount of training—if it ever transitions into a habit. The attentive reader might have noted that our contingency degradation test required participants to withhold an overtrained response, precisely the type of behavior that appears resistant to habitual control. This aligns with Du and Haith's hypothesis [47], providing a plausible explanation for our difficulty in detecting habit expression in the overtraining condition.

Even when habit and goal-directed systems compete, human participants typically adapt their responses according to goal-directed processing, regardless of the amount of prior training. Studies have shown that impairing goal-directed control is often necessary to reveal habit-driven behavior [44,45,47,48]. A successful approach involves limiting response preparation time [44,45,47]. For example, Hardwick et al. [44] used a forced-response test with new stimulus-response mappings following extensive training on a different, partially incongruent configuration. Habitual errors increased when participants had only 300–600 ms to prepare their response, but not with longer preparation times. Similarly, Luque et al. [45] found that, after three days of operant training, participants successfully selected goal-directed responses during an outcome devaluation test. However, under time pressure (≤500 ms), switch costs were higher after overtraining, indicating competing responses. This interference disappeared when the response time was extended to 4 seconds [45]; see also [21,49].

Cognitive control is impaired by stress. Experimentally induced stress has been suggested to facilitate the shift from goal-directed control toward more rigid habits when using outcome devaluation procedures after instrumental learning [48,50] (see also [51], for similar results with contingency degradation). EEG data from Meier et al. [52] indicate that stress might enhance stimulus-response processing while weakening outcome representations. However, most studies on stress and habit formation rely on single-session, two-choice response tasks, which may overestimate habit effects by failing to distinguish them from goal-directed errors. This limitation could be addressed by including 'minimal training' control conditions, where goal-directed errors occur but habits have not yet formed. Such an approach could provide an interesting path for future research on the effects of stress on habit induction. Other recent approaches exploring overtraining effects

in humans are those employing a mobile-phone app methodology [53,54], trying to detect habit expression in a more naturalistic environment. This could be a promising path; however, more research is needed to corroborate if these procedures are robust tools for indexing habit formation (for reliability test-retest studies of Hardwick and Luque tasks see [55]).

Taken together, these findings suggest that uncovering habitual behavior requires impairing goal-directed control while also ensuring that one condition involves significantly more training than another to distinguish habitual from goal-directed responses. Future studies using a contingency degradation paradigm could enhance test sensitivity by encouraging rapid responses, such as offering reward bonuses for faster performance during training and testing or imposing time constraints on responses.

Difficulties in translating animal models to human behavior are common in behavioral and neuroscience research. In some cases, this is due to methodological issues in animal studies, such as small sample sizes or the failure to report sufficient procedural details, which hinder the replication of these effects [56]. However, we do not believe this to be the case in habit research. Since Adams' [4] seminal work, animal studies have consistently shown that the transition from goal-directed control to habitual behavior depends on the amount of training, as demonstrated through outcome devaluation and contingency degradation tests [5,16]. These effects have been widely replicated [6,43,57]. Additionally, lesion studies in rats have identified distinct neural circuits underlying habitual and goal-directed actions (for reviews, see [3,58]). Why, then, is it so difficult to observe overtraining effects in humans? The answer likely lies in the need to refine experimental parameters to optimize habit detection in human participants. While we are making progress in developing better tools for studying habits, the precise conditions that facilitate habit expression in humans remain unclear. In our protocol, we kept a constant positive response-outcome contingency to facilitate the development of persistent responses, before introducing a change in contingency relationship, sharing that aspect of our design with classical animal studies which reported overtraining effects [5,16]. However, participants had no problem adjusting their response despite the amount of training. Interestingly, even in animal research, methods are still evolving, with improved protocols emerging to enhance the study of habitual behavior (e.g., [59,60]).

As part of this effort, the current experiments aimed to evaluate the validity of response rates during contingency degradation as a measure of habit expression in humans. This type of research is crucial for the habit learning field, especially as new tools for studying habits continue to emerge. Before assuming that a measure reliably captures habitual behavior, it must first be validated. This precaution is particularly important when the measure analyzes individual differences, such as in studies involving special populations. Without proper validation, researchers risk falling into a circular explanation. For example, one might use a contingency degradation test to study habits in a population, such as OCD patients, under the assumption that they have an altered habit system. Then, the validity of the test as a measure of habit expression is inferred from the observed differences between OCD patients and healthy controls.

In this study, we used the amount of training to investigate whether a measure truly captures habit expression. This approach has been highly effective in identifying factors and methodologies that facilitate habit detection. So far, evidence indicates that the best strategies for impairing goal-directed control and allowing habits to emerge involve forcing quick responses [44] or analyzing response time costs [45]. Stress-induction procedures and clinical studies (which assume that certain populations may have a predisposition toward habit formation) may also offer insights into habitual behavior. However, in our view, these studies should not bypass validation efforts—such as systematically manipulating the amount of training—to confirm that their measures are genuinely indexing habits.

In conclusion, observing habits in human research remains a challenge, particularly when using classical procedures such as outcome devaluation and contingency degradation tests. Unlike in animal research, there is no clear evidence that humans become insensitive to contingency degradation after overtraining an instrumental response. This calls into question the utility of this test for characterizing habit formation and expression in humans. Moving forward, a collective effort is needed to refine experimental protocols while rigorously validating them. The current research contributes to this effort by reinforcing the importance of training manipulations as a means of evaluating habit measures.

## Supporting information

**S1 File. Additional supporting information, tables and figures.**
(DOCX)

## Author contributions

**Conceptualization:** Joaquín Morís, Pedro L. Cobos, Francisco J. López, David Luque.

**Data curation:** Pablo Martínez-López, Joaquín Morís.

**Formal analysis:** Pablo Martínez-López, Joaquín Morís.

**Funding acquisition:** David Luque.

**Investigation:** Sara Molinero, Pablo Martínez-López, María J. Quintero.

**Methodology:** Sara Molinero, Pablo Martínez-López, Joaquín Morís, Pedro L. Cobos, Francisco J. López, David Luque.

**Project administration:** David Luque.

**Resources:** David Luque.

**Software:** Pablo Martínez-López, Joaquín Morís.

**Supervision:** David Luque.

**Validation:** Sara Molinero, Pablo Martínez-López, Joaquín Morís.

**Visualization:** Pablo Martínez-López.

**Writing – original draft:** Sara Molinero.

**Writing – review & editing:** Pablo Martínez-López, Joaquín Morís, María J. Quintero, Pedro L. Cobos, Francisco J. López, David Luque.

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
