## [Decision Letter · Decision Letter 0]

30 Jun 2025

The Degraded Contingency Test Fails to Detect Habit Induction in Humans

PLOS ONE

Dear Dr. Molinero Infante,

Thank you for submitting your manuscript to PLOS ONE. I am very grateful to the two reviewers who took the time to read and comment on the manuscript. As you can see, both reviewers thought the manuscript (and studies) were excellent, but were very consistent in noting that there is an issue relating to a potential overtraining effect in the standard (undertrained group), which may affect interpretation of the results. It would be good if you could address these points in a revision.

We look forward to receiving your revised manuscript.

Kind regards,

Poppy Watson

Academic Editor

PLOS ONE

Journal Requirements:

[This work was supported by grant PID2021-126767NB-I00 (S.M, D.L.), Ministry of Science, Innovation and Universities (MICIU); and grant PRE2022-103151 (P.M.L.), MICIU/ESF.].

3. Thank you for stating the following in your manuscript:

[This work was supported by grant PID2021-126767NB-I00 (S.M, D.L.), Ministry of Science, Innovation and Universities (MICIU); and grant PRE2022-103151 (P.M.L.), MICIU/ESF.]

[This work was supported by grant PID2021-126767NB-I00 (S.M, D.L.), Ministry of Science, Innovation and Universities (MICIU); and grant PRE2022-103151 (P.M.L.), MICIU/ESF.]

4. Please amend the manuscript submission data (via Edit Submission) to include author Sara Molinero.

5. Please amend your authorship list in your manuscript file to include author Sara Molinero Infante.

6. Please include captions for your Supporting Information files at the end of your manuscript, and update any in-text citations to match accordingly. Please see our Supporting Information guidelines for more information: http://journals.plos.org/plosone/s/supporting-information .

Reviewers' comments:

Reviewer's Responses to Questions

**Comments to the Author**

1. Is the manuscript technically sound, and do the data support the conclusions?

Reviewer #1: Yes

Reviewer #2: Partly

2. Has the statistical analysis been performed appropriately and rigorously?

Reviewer #1: Yes

Reviewer #2: Yes

3. Have the authors made all data underlying the findings in their manuscript fully available?

Reviewer #1: Yes

Reviewer #2: Yes

4. Is the manuscript presented in an intelligible fashion and written in standard English?

Reviewer #1: Yes

Reviewer #2: Yes

Reviewer #1: This manuscript reports three pre-registered experiments examining whether a contingency degradation test can detect habit formation in humans as a function of overtraining. Across studies, participants were trained under either standard or overtrained conditions and then tested under degraded contingencies. The authors found no evidence that overtraining led to increased habitual responding. The paper is detailed, well-written, and methodologically rigorous, with strong open-science practices including pre-registration, data/code sharing, and clear reporting.

Overall, I find this paper thorough and thoughtful paper. The introduction is well-grounded in the literature, and the discussion is comprehensive, raising many relevant points about methodological limitations in habit research and the challenges in translating animal findings to humans.

Most of my comments are minor (see below). The main point I would like to raise concerns the interpretation of the results:

The authors interpret the null effect of overtraining as evidence that participants remained goal-directed across conditions. However, from Figure 2, it appears that participants show fairly habitual behavior even after short training - particularly in Experiment 2, where performance under degraded contingencies seems completely habitual (ratio score around 0.5). This raises the possibility that both standard and overtrained cues in the task elicited habitual behavior. The null effect may actually reflect generalization of the learning/performance/habit to the standard cue (which seems reasonable in the current task in my opinion).

Additionally, another observation that may challenge the authors’ interpretation (and could benefit from further reflection in the paper) is the apparent bimodal distribution in response tendencies. That is, some participants appear to become habitual while others remain sensitive to contingency degradation, regardless of training duration.

Minor points:

1. The use of the word “unaffected” in the abstract may be too strong. A moderate reduction in behavior in degraded blocks may still reflect habitual behavior to some degree. Consider softening this claim to better reflect the nuance of the data.

2. The last paragraph in the Introduction speaks about the (series of) experiments in a way that feels they have been introduced before (while they were not). Consider rephrasing accordingly.

3. Page 5: More details on the power analysis for Experiment 1 would be helpful.

4. Table 1: The structure is confusing. The rows appear as labeled with cue types (one the left), which they are not. Consider revising the format or separating the cue labels clearly (e.g., as a separate legend or in the caption). Additionally, it says: “positive/negative or habit test” but “negative” is the habit test in this paradigm. Please revise this for clarity.

5. Page 8: “This was confirmed by a correlation analysis between programmed delta P and experienced delta P, r = .867, p < .001. Causality ratings further confirmed that participants were aware of contingency shifts, correctly identifying changes from positive to negative contingencies.” It is not clear if this refers to two separate analyses. Please clarify with separate reporting if so.

6. Page 9: “participants who completed only the first habit test...” is a bit misleading. As I understand it, all participants completed both tests; this is just a comparison of the first test block between groups. Please clarify.

7. Page 16: The claim that more extensive training would not have changed the result rests on weak evidence. The authors cite asymptotic response rates and prior studies, but:

- Reaching asymptote doesn’t preclude additional habit formation.

- Other work (e.g., Lally et al., 2010) shows habits may take a long time to form.

- Luque et al. (2020) used a different paradigm and test, so its relevance here is limited.

That said, this is the authors' interpretation, and I do not think it necessarily needs to be removed or changes. However, I do encourage them to reconsider this point.

8. In discussion, when discussing more effective methods of revealing habit (e.g., time pressure, stress), the authors could mention Gera et al. (2024), who recently demonstrated habit formation over extended training using a novel smartphone-based task. This would enrich the discussion of methodological innovation and bridge lab-to-real-world settings.

9. When discussing the challenges associated with outcome devaluation procedures, overtraining effects, and identifying effective methods for inducing habits, it would be worthwhile to mention recent alternative approaches, such as Gera et al. (2024) who demonstrated habit formation (devaluation insensitivity) as a function of training duration using a smartphone app paradigm that allows for more extended training.

Other Thoughts:

1. I wonder whether excluding the first few seconds of the test blocks (where participants might learn about the contingency change) affects the results.

2. The task structure/components likely prompts constant vigilance to contingency, potentially undermining habit expression.

Reviewer #2: This manuscript investigates whether overtraining a stimulus-response-outcome association enhances habit formation, as measured by the degraded contingency test. Across three preregistered, sufficiently-powered studies, the authors find that overtraining did not lead to greater insensitivity to outcome degradation compared to a standard training. This null result is robustly supported by Bayesian analyses. The authors interpret this as a potential failure of the degraded contingency test to adequately measure habit induction in this context. The study is well-designed and the findings are a valuable contribution to the literature on habit formation. My comments below are intended to help strengthen the interpretation of the results and improve clarity in a few key areas. Please consider the following as suggestions. If I’ve misunderstood any aspect or you disagree with a point, a brief clarification is more than sufficient; there is no obligation to revise the manuscript based on these comments.

Major

My main points relate to the interpretation of the results as showing no habit induction.

1. The authors convincingly demonstrate that there is no difference in outcome degradation sensitivity between the overtrained and standard conditions. However, the conclusion that this represents a failure to induce habits is open to debate. Looking at the data presented (particularly in Figure 2), a notable proportion of participants in both conditions exhibit response ratios near 0.5, which is indicative of habitual responding. This pattern is especially apparent in the between-subject comparisons, which arguably provide a cleaner measure. This raises an important alternative interpretation: rather than a failure to induce habits via overtraining, it's possible that the "standard" training protocol—conducted over three days—was already sufficient to induce robust habits in many participants. If this were the case, it would lead to a floor effect, where the overtraining has no additional effect to be measured. I would encourage the authors to discuss this possibility more thoroughly. Could the conclusion be reframed, not as a failure of the test, but as evidence that even moderate training can be sufficient to establish habitual control under these conditions?

• The manuscript frames the findings primarily as a shortcoming of the degraded contingency test itself. This is one possibility, but I would encourage the authors to more thoroughly consider an alternative framing. While the results demonstrate failures to demonstrate habit induction with overtraining effect, this may also be a consequence of the specific experimental parameters chosen (e.g. Schreiner et al 2019). As subtle changes in experimental design can have significant effects on the expression of habitual behavior, a more cautious interpretation may be warranted. It would therefore strengthen the paper to more explicitly balance the discussion between the test being the issue versus the specific training conditions being the reason for the observed null effect. The authors mention several elements of the study design that may contribute, e.g. that habits were tested as withholding an overtrained response (which may not become habitual) and stable vs changing contingency relationships and compare this with other human studies. But it’s less clear how the current study compares with the rodent literature where overtraining effects on contingency degradation tasks are reliably found. How do the key parameters of the current study (stable vs. varying contingency relationships; specific ΔP values for the degradation phase) compare to those classic animal studies? This could help contextualize the current results and strengthen the argument regarding the critical importance of specific experimental parameters in observing habit formation.

Minor

Methods:

• The response ratio is (appropriately) used to obtain a relative measure of goal-directedness. However, since it would be very insightful to see (individual differences in) the absolute number of responses participants made. Would it be possible to include a (supplementary) plot showing the absolute number of responses for each cue type over test blocks? This would provides more insight into the behavior of participants to readers.

• The manuscript includes both within- and between-subject comparisons, but the rationale and description are not clearly articulated in the Methods section of Study 1. I found it difficult to understand at first.

• The speed of data collection for Experiments 1 and 2 is remarkable, with each study seemingly completed within three days. This implies a very high throughput of participants. A brief sentence in the Methods explaining the logistics (e.g., use of multiple testing stations, an online platform) would be helpful and would mitigate potential reader questions about feasibility.

Experiment 1:

• When reporting the power analysis, could the authors specify the target effect size and the alpha level that determined a sample size of 50 participants would be sufficient?

• The term "habit test" seems to appear for the first time in a table. It would improve clarity to introduce and define this term in the main text when the procedure is first described.

Experiment 2:

• Could the authors please provide the rationale for including 57 participants in this experiment?

Experiment 3:

• The justification for Experiment 3 ("an exploratory analysis suggested a trend") is somewhat vague. It would help the reader if the authors could elaborate on the nature of this trend and the specific hypothesis that motivated this follow-up study.

• In the Results section, evidence for overtrained vs standard cue was interpreted as strong evidence, while the BF01 was only 2.63. Seems like a small mistake.

**Do you want your identity to be public for this peer review?** For information about this choice, including consent withdrawal, please see our Privacy Policy

Reviewer #1: No

Reviewer #2: No

---

## [Author Response · Author response to Decision Letter 1]

15 Sep 2025

14th September 2025

Dear Dr. Watson,

Thank you for your considered response to our manuscript ‘The degraded contingency test fails to detect habit induction in humans’. We have closely considered the changes you and your reviewers have suggested and have responded to each point below.

We also performed new analysis that hopefully it would help to support the interpretation of the main results.

We hope that you will now find this revised version of the manuscript suitable for publication in Plos One.

Kind regards,

Sara Molinero

(on behalf of the authors)

---

## [Editor Report · Decision Letter 1]

23 Sep 2025

The Degraded Contingency Test Fails to Detect Habit Induction in Humans

PONE-D-25-19259R1

Dear Dr. Molinero,

We’re pleased to inform you that your manuscript has been judged scientifically suitable for publication and will be formally accepted for publication once it meets all outstanding technical requirements.

Kind regards,

Poppy Watson

Academic Editor

PLOS ONE
---

## [Editor Report · Acceptance letter]

PONE-D-25-19259R1

PLOS ONE

Dear Dr. Molinero,

I'm pleased to inform you that your manuscript has been deemed suitable for publication in PLOS ONE. Congratulations! Your manuscript is now being handed over to our production team.

Kind regards,

on behalf of

Dr. Poppy Watson

Academic Editor

PLOS ONE